# Native metabolomics identifies the rivulariapeptolide family of protease inhibitors

Raphael Reher[1,2,3], Allegra T. Aron[4], Pavla Fajtová[4], Paolo Stincone[5], Berenike Wagner[5,6], Alicia I. Pérez-Lorente[7], Chenxi Liu[4], Ido Y. Ben Shalom [4], Wout Bittremieux [4], Mingxun Wang[4], Kyowon Jeong [8], Marie L. Matos-Hernandez [9], Kelsey L. Alexander[1,10], Eduardo J. Caro-Diaz [9], C. Benjamin Naman [11], J. H. William Scanlan[12], Phil M. M. Hochban[12], Wibke E. Diederich[12], Carlos Molina-Santiago [7], Diego Romero [7], Khaled A. Selim [5,6], Peter Sass [5,6], Heike Brötz-Oesterhelt [5,6,13], Chambers C. Hughes [5,6,13], Pieter C. Dorrestein [4], Anthony J. O'Donoghue[4], William H. Gerwick [1,4] ✉ & Daniel Petras [1,4,5,6] ✉

The identity and biological activity of most metabolites still remain unknown. A bottleneck in the exploration of metabolite structures and pharmaceutical activities is the compound purification needed for bioactivity assignments and downstream structure elucidation. To enable bioactivity-focused compound identification from complex mixtures, we develop a scalable native metabolomics approach that integrates non-targeted liquid chromatography tandem mass spectrometry and detection of protein binding via native mass spectrometry. A native metabolomics screen for protease inhibitors from an environmental cyanobacteria community reveals 30 chymotrypsin-binding cyclodepsipeptides. Guided by the native metabolomics results, we select and purify five of these compounds for full structure elucidation via tandem mass spectrometry, chemical derivatization, and nuclear magnetic resonance spectroscopy as well as evaluation of their biological activities. These results identify rivulariapeptolides as a family of serine protease inhibitors with nanomolar potency, highlighting native metabolomics as a promising approach for drug discovery, chemical ecology, and chemical biology studies.

[1]Scripps Institution of Oceanography, University of California San Diego, La Jolla, CA, USA. [2]Institute of Pharmacy, Martin-Luther-University Halle-Wittenberg, Halle, Germany. [3]Institute of Pharmaceutical Biology and Biotechnology, University of Marburg, Marburg, Germany. [4]Skaggs School of Pharmacy and Pharmaceutical Science, University of California San Diego, La Jolla, CA, USA. [5]Cluster of Excellence "Controlling Microbes to Fight Infections" (CMFI), University of Tuebingen, Tuebingen, Germany. [6]Interfaculty Institute of Microbiology and Infection Medicine, University of Tuebingen, Tuebingen, Germany. [7]Instituto de Hortofruticultura Subtropical y Mediterránea "La Mayora," Consejo Superior de Investigaciones Científicas, Departamento de Microbiología, Universidad de Málaga, Málaga, Spain. [8]Applied Bioinformatics, Computer Science Department, University of Tuebingen, Tuebingen, Germany. [9]Department of Pharmaceutical Sciences, School of Pharmacy, University of Puerto Rico - Medical Sciences Campus, San Juan, Puerto Rico. [10]Department of Chemistry and Biochemistry, University of California San Diego, La Jolla, CA, USA. [11]Li Dak Sum Yip Yio Chin Kenneth Li Marine Biopharmaceutical Research Center, Department of Marine Pharmacy, College of Food and Pharmaceutical Sciences, Ningbo University, Ningbo, China. [12]Department of Pharmaceutical Chemistry and Center for Tumor Biology and Immunology (ZTI), University of Marburg, Marburg, Germany. [13]German Center for Infection Research, Partner Site Tuebingen, Tuebingen, Germany. ✉e-mail: wgerwick@health.ucsd.edu; daniel.petras@uni-tuebingen.de

Specialized metabolites, often referred to as natural products, are a tremendous pool of chemically diverse and pharmaceutically active organic compounds. By some estimates, more than 50% of all current pharmaceuticals are based on or inspired by natural products[1]. Nevertheless, the vast majority of biological activities and pharmaceutical potential of specialized metabolites, as well as their ecological functions, still remain to be discovered[2].

While mining genome and metagenome data has begun to provide an overview of the biosynthetic potential of nature[3,4], most specialized metabolites remain inaccessible, as the living organism that produces these metabolites cannot be cultured or their gene clusters remain silent under laboratory culture conditions. Natural product discovery and chemical ecology studies in environmental systems are hence becoming more and more attractive[5]. Along with next-generation sequencing technologies, recent instrument and computational advances in nuclear magnetic resonance (NMR) spectroscopy and non-targeted liquid chromatography-tandem mass spectrometry (LC-MS/MS) offer tremendous assistance in exploring the uncharted metabolic space of nature[6]. These tools enable large-scale compound dereplication, rapid identification of chemical analogs, and de novo annotation of molecular formulas, substructures, chemical classes, and structures[7–12]. However, the assignment of bioactivity to newly identified metabolites typically requires assays using pure compounds. Therefore, the isolation of specialized metabolites is typically guided by repetitive bioactivity assays. Together with full structure elucidation, this process usually takes months and is, therefore, a major bottleneck for the systematic exploration of nature for novel pharmaceutically active compounds and comprehensive chemical ecology studies.

A logical next step is the development of activity metabolomics[13] or functional-metabolomics[14] approaches that aim to add functional information to the metabolites detected in a given system. Native electrospray ionization (ESI) such as immobilized enzyme MS (IEMS)[15] and affinity mass spectrometry (MS) such as pulsed ultrafiltration (UF) MS, size exclusion (SEC) or affinity bead-based pull-down assays are increasingly being used to analyze non-covalent binding of biomolecules[16–21]. An important difference between native MS and affinity MS is that native MS detects ligands directly bound to a protein, whereas affinity MS approaches typically measure ligand binding indirectly as free compounds. For affinity MS, the target protein is captured by UF, SEC, centrifugation, or magnetic removal and the released ligand is subsequently identified by small molecule MS analysis[22–25]. Both native and affinity MS approaches have been applied with single compounds as well as substrate pools, which allows for the simultaneous screening of complex natural compound libraries[26–28]. Depending on the experimental setup, an important limitation in the use of ligand pools is that multiple ligands compete for binding of the target at the same site, and therefore compounds with the highest affinity or concentration are more easily discovered. However, this limitation can be overcome by using a molar excess of protein compared with the total molar concentration of the ligand library[15]. Additionally, the annotation of bound ligands remains a challenging task, especially if the ligand pool contains multiple isobaric compounds. While direct infusion native MS workflows have been developed that can identify metabolites bound to proteins by MS/MS[29], combining the separation power of ultra-high-performance liquid chromatography (UHPLC) and the selectivity of native MS and MS/MS would offer a promising improvement to decipher protein-metabolite interactions out of complex biological mixtures, such as environmental samples. However, typical UHPLC mobile phase conditions disfavor non-covalent protein binding due to an acidic pH and high organic solvent content. To perform native MS coupled to UHPLC, we developed an experimental setup that increases pH and water content of the mobile phase post-column and infuses a protein binding partner before entering the ESI interface (Fig. 1). As the protein is constantly infused post-column, one can monitor the intact protein mass over the LC-MS run and observe mass shifts when eluting metabolites bind to the protein at a defined retention time. Using collision-induced dissociation (e.g., Higher Energy C-Trap Dissociation (HCD) in the setup), the complex can be dissociated again in the mass spectrometer, and a "binding threshold" can be applied to distinguish between specific and nonspecific binding. In combination with parallel non-targeted MS/MS analyses, the mass and compound ID or compound class can be assigned (level 2 or level 3 annotation[30,31]).

Cyanobacteria are known to be a rich source of highly bioactive natural products[32–35], and in particular, protease inhibitors from numerous chemical classes[36–41]. Protease inhibitors are key compounds used for the treatment of viral infections (SARS-CoV-2, HIV, and Hepatitis C)[42,43], cancer[44], diabetes[45], hypertension[46], and as general anticoagulants[47]. Several of the approved protease inhibitors are analogs of natural products such as aliskiren, captopril, and carfilzomib that target renin, angiotensin-converting enzyme, and the proteasome, respectively[48].

In this study, we screen for protease inhibitors from an environmental cyanobacteria biofilm as an application of the native metabolomics workflow. We use chymotrypsin as the protease target to identify inhibitors from a marine cyanobacteria community. We identify 30 chymotrypsin binders in the methanolic crude extract with a single LC-MS run. The masses and MS/MS spectra of the binders are then queried against structural and spectral databases, revealing that most of them are unknown. This leads to the targeted isolation and structure elucidation of a family of highly potent protease inhibitors, which we termed rivulariapeptolides.

## Results

### Development of the native metabolomics approach

In a crude extract, native metabolomics provides binding information about each compound towards a protein of interest. In the experimental setup, we utilized a single 10-min LC-MS run to discover compounds that bind to the serine protease, chymotrypsin. The workflow is as follows: a crude extract is analyzed using native ESI while the protein of interest is infused post-column throughout the entire LC gradient. Binding of a small molecule to the protein of interest results in a peak with a mass corresponding to the protein bound to the compound. The $m/z$ difference between the protein-ligand complex and the unbound protein reveals the molecular weight of the ligand while the ratio of the intensity of the protein-ligand peaks relative to the unbound protein peaks hints toward the relative binding affinity under the given conditions. To reduce nonspecific binding, native metabolomics experiments were performed in all-ion fragmentation (AIF) mode, using a threshold collision energy of 20% in the HCD cell.

We first optimized the pH for the native mass spectrometric acquisition of chymotrypsin and confirmed that the enzyme remains active under the native metabolomics buffer conditions. As a positive control for binding, we used molassamide[49], a known non-covalent serine protease inhibitor of the 3-amino-6-hydroxy-2-piperidone (Ahp)-cyclodepsipeptide family. We found that an ammonium acetate solution of pH 4.5 showed the highest peak intensity (Supplementary Fig. 1a). Volatile ammonium acetate solution is the most commonly used non-denaturing solvent in native MS measurements of proteins. Even though having initially neutral pH, ammonium acetate has no buffering capacity at pH 7, and in positive ion mode, there are several factors that tend to acidify the analyte solution. In the presence of ammonium acetate, the pH may drop to values as low as $4.75 \pm 1$ in the final ESI droplets, reflecting the pKa of acetate buffer[50]. Next, we injected a serial dilution of molassamide into the native metabolomics LC-MS setup, where it was mixed post-column with a constant concentration of chymotrypsin. The protein-ligand complex was detected at a deconvoluted mass of 26195.1 Da and the unbound apoprotein at 25232.6 Da (Supplementary Fig. 1b). The observed Δ mass of 962.5 Da

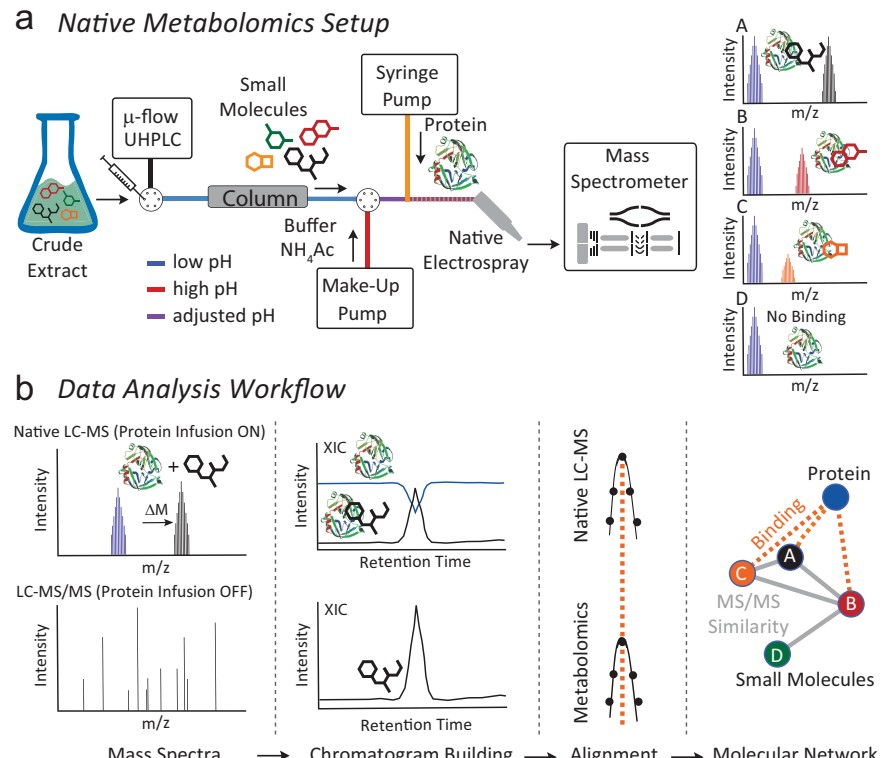

**Fig. 1 | Overview of the native metabolomics workflow. a** Native metabolomics setup. A crude extract is separated by μ-flow UHPLC. The pH is adjusted after chromatography with ammonium acetate to native-like conditions via the make-up pump. Orthogonally, the protein of interest is infused, and the resulting protein-binder complexes are measured by MS. The procedure is repeated as a metabolomics run (high-resolution UHPLC-MS/MS acquisition without protein infusion). For the data analysis (**b**) the Δ *m/z* and retention time of the native MS run are correlated with *m/z* values and retention time of the metabolomics (LC-MS/MS) run and subsequently visualized using molecular networking and retention time pairing that links the observed mass differences of the protein in the native state vs bound states with the parent mass of MS/MS spectra of the small molecule.

matches the mass of molassamide (962.4749 Da). We also observed a minor peak with a mass shift that corresponds to two molassamide molecules in the multiple charged spectrum (Supplementary Fig. 1d), which can be an indication of nonspecific binding[51]. However, as molassamide is a known nanomolar inhibitor, we cannot rule out whether the second peak could be due to nonspecific binding effects or if it represents a secondary weaker interaction. After deconvolution and integrating the peak area of the protein-ligand complex and plotting it against the molassamide concentration in the peak window, we obtained a binding curve. The resulting curve depicts a concentration-dependent increase of protein-ligand to unbound protein ratio with increasing ligand concentration (Supplementary Fig. 1c). The limit of detection for the molassamide-chymotrypsin interaction was between 0.1 and 1 μg/mL (1 – 10 ng on column) (Supplementary Fig. 1e). To further test the biological relevance of the native metabolomics conditions, chymotrypsin was assayed with a crude extract from the cyanobacterium *Rivularia* sp. using a fluorescence substrate competition assay. The bioassay conditions were designed to mimic the pH and solvent composition expected in the native mass spectrometry setup. Although chymotrypsin is optimally active in near-neutral pH, it retains activity in 10 mM NH₄Acetate at pH 4.5. Under these conditions, chymotrypsin was completely inhibited by 10 μg/mL of the extract with 50% inhibition at 0.84 μg/mL (Supplementary Fig. 1f). Chymotrypsin was then assayed in increasing concentrations of acetonitrile (ACN) to determine if enzyme activity was retained in this solvent. The activity was reduced by 9 to 34% in ACN concentrations up to 33.3% v/v. In the presence of 41.7% v/v ACN, which corresponds to the end of the UHPLC gradient after the make-up addition, activity was decreased by 70% (Supplementary Fig. 1g). These results confirmed that chymotrypsin can be used as a target protease for

native metabolomics as it retains activity at pH 4.5 in ACN concentrations up to 41.7% and binds to compounds from an inhibitory crude extract. The NH₄Acetate concentration can have a large influence on the ionization and dissociation constants of non-covalent protein-binder complexes. Interestingly, higher NH₄Acetate concentration can lead to both higher or lower binding affinities depending on the protein-ligand pair investigated[52]. We tested a series of NH₄Acetate concentrations (10–100 mM, Supplementary Fig. 2) and found 10 mM to be the most favorable for studying chymotrypsin-molassamide complexes (highest relative abundance, highest chymotrypsin-molassamide to chymotrypsin ratio, lowest abundance of interfering NH₄Acetate clusters). To test the variability of the changing ACN concentration during the LC separation, we performed a series of flow injections (without column) over the full gradient (Supplementary Fig. 3a). The XIC of the molassamide bound chymotrypsin reveals similar signal responses throughout the gradient (5-99% ACN on column). While injecting a pool of control compounds at concentrations of 10 μg/mL, we observed compound-specific signal responses of the chymotrypsin-ligand complexes (Supplementary Fig. 3b). To further assess the specificity of the protein-small molecule interactions, we evaluated the binding of the linear oligopeptidic cysteine protease inhibitor gallinamide A, the cyclic depsipeptide FR900359, the isoflavone genistein, the phenol phloroglucinol, and the anthraquinone quinalizarin[34,53,54]. We did not observe the binding of these negative controls to chymotrypsin under native MS conditions (Supplementary Fig. 3c).

## Native metabolomics reveals chymotrypsin binders

Following the successful proof-of-concept experiments, we next screened for potential chymotrypsin binders from a crude extract of a

biofilm from the marine cyanobacterium *Rivularia* sp, collected from coral sediments at Carlos Rosario Beach in Culebra, Puerto Rico, U.S. The methanol extract was separated by reversed-phase UHPLC and ammonium acetate buffer and chymotrypsin were infused post-column, prior to native ESI and acquisition of mass spectrometry data in the high *m/z* range (2500-5000 *m/z*). The crude extract was subsequently re-injected, without the infusion of chymotrypsin to obtain high-resolution LC-MS/MS data of compounds in the extract in the low *m/z* range (300−2000 *m/z*).

As the first step of data analysis, we plotted the total ion current (TIC) of the crude extract metabolomics (Fig. 2a, left) and extracted ion chromatogram (XIC) of the apo-chymotrypsin from charge state deconvoluted native mass spectrometry data (Fig. 2a, right) which shows several negative peaks in the range of 4.5–5.5 min. The decrease in a signal of the apoprotein in that retention time range is due to the emergence of larger masses that correspond to protein-ligand complexes. After feature finding of the deconvoluted masses and matching of the parallel metabolomics LC-MS/MS data of the crude extracts by retention time and exact mass matching, we could identify more than 30 potential small molecule-protein complexes. To display the family of small molecules that form protein-ligand complexes and to show their structural relations, we visualized them in a correlation molecular network (Fig. 2b and Supplementary Fig. 4) that is based on their MS/MS similarity (gray line), retention time, and mass matching between protein and small molecules through the red dashed lines.

Two of the most abundant chymotrypsin binders have *m/z* 1186.6400 and *m/z* 1156.5923 that also show a perfect overlap of the chromatographic profiles between intact protein and metabolomics LC-MS/MS data (Fig. 2c and Supplementary Fig. 5). Based on the high relative abundance we targeted them for further purification, structure determination by NMR and orthogonal protease inhibition assays.

## Rivulariapeptolides, a family of new Ahp-cyclodepsipeptides

The potential chymotrypsin-binder with the *m/z* 1186.6400, identified by native metabolomics, was next targeted for isolation and structure elucidation, using state-of-the-art high-resolution MS/MS and NMR approaches[55] (see also Supplementary Methods). We first separated the *Rivularia* crude extract into four fractions of decreasing polarity via solid phase extraction. SMART NMR[11] analysis was applied to the most hydrophilic fraction and all but one structure of the top 10 SMART results were predicted as cyclic depsipeptides (Fig. 3a, b and Supplementary Fig. 6), including 6 of the top 10 as Ahp-cyclodepsipeptides (three from marine filamentous cyanobacteria: somamide B, molassamide, lyngbyastatin 6)[49,56–58]. Complementarily, MS/MS-based molecular networking analysis of the *Rivularia* crude fractions assisted with the annotation of the known Ahp-peptides molassamide, kurahamide, and loggerpeptin A, along with several putatively new ones (Fig. 3a, b and Supplementary Fig. 7a)[49,57,59]. Next, the SIRIUS and ZODIAC tools[10,60] were applied to determine the molecular formula of the chymotrypsin-binding feature, with exact mass *m/z* 1186.6400 [M + H]⁺, as $C_{61}H_{87}N_9O_{15}$ (0.5 ppm). Subsequently, we classified the MS/MS spectrum as indicative of a "cyclic depsipeptide" based on the classification with CANOPUS[9]. Further substructures of the molecule were predicted as benzene, hydroxy-benzene, and proline/*N*-acyl-pyrrolidine derivatives, as well as piperidinone/delta-lactam for the Ahp-family defining moiety (Fig. 3c and Supplementary Fig. 7b).

To unambiguously determine the structure, we isolated *m/z* 1186.6400, named rivulariapeptolide 1185 (1), and performed 1D/2D NMR experiments and manual MS/MS interpretation (Fig. 3d, Supplementary Figs. 8–13, 43, and Supplementary Table 1). Subsequently, we targeted the isolation of further rivulariapeptolides by preparative HPLC, based on their protein-ligand complex ratios from the native metabolomics experiments as well as their relative abundance. In that way, we isolated and elucidated the planar structures of the

rivulariapeptolides 1185, 1155, 1121, and 989 (1, 2, 3, 4) with the exact masses 1186.6400 [M + H]⁺ ($C_{61}H_{88}N_9O_{15}$, 0.5 ppm), 1156.5923 [M + H]⁺ ($C_{59}H_{82}N_9O_{15}$, − 0.2 ppm), 1122.6080 [M + H]⁺ ($C_{56}H_{84}N_9O_{15}$, − 0.1 ppm) and 989.4978 [M + H]⁺ ($C_{50}H_{69}N_8O_{13}$, − 0.1 ppm). In addition to rivulariapeptolides, we identified a new molassamide derivative that we termed "molassamide B" (6) with *m/z* 1041.3924 [M + H]⁺ ($C_{48}H_{66}BrN_8O_{13}$, −0.3 ppm), which is ortho-brominated (Supplementary Figs. 14–43 and Supplementary Tables 2, 3). The absolute configurations of the amino acids were determined by UHPLC-MS analysis of the acid hydrolysates of 2 and its pyridinium dichromate oxidation product and subsequent advanced Marfey's analysis (Supplementary Fig. 44a). The analyses revealed L-configurations for all amino acids as is the case for other cyanobacterial Ahp-cyclodepsipeptides. The relative configuration of the stereocenters of the (3 *S*, 6 *R*)-Ahp-unit as well as the geometry of the double bond (*Z*-configuration) of the 2-amino-2-butenoic acid (Abu) moiety was determined by NOESY and HMBC NMR experiments (Supplementary Fig. 44b). Assuming that the other rivulariapeptolides (1, 3, 4) and molassamides (5 + 6) described here originate from the same biosynthetic nonribosomal peptide synthetase, they most probably also share the same backbone configuration. The highly comparable MS/MS and NMR data sets of compounds 1-6 (Supplementary Figs. 8–43 and Supplementary Tables 1–4) provide additional evidence that the discovered Ahp-cyclodepsipeptides from this study share the same configuration.

Finally, the chymotrypsin inhibitory activities of the purified Ahp-cyclodepsipeptides 1-6 were assessed by specific biochemical assays and confirmed the results of the native metabolomic protein infusion MS experiments (Fig. 4). All six compounds were found to be nanomolar chymotrypsin inhibitors with compound 1 being the most potent (Fig. 4, IC₅₀ = 13.17 nM). The newly described family of rivulariapeptolides is characterized by a rare (duplicated) *N*-butyrylated proline moiety in the side chain. Peptides 1-4 and 6 are among the six most potent natural chymotrypsin inhibitors from the compound class of Ahp-cyclodepsipeptides that have been reported to date[36] (Supplementary Table 5).

Intriguingly, a single ortho-bromination in the *N*-methyltyrosine moiety led to a 35-fold increase in potency for the new compound 6 (IC₅₀ = 24.65 nM), when compared to the known compound 5 (IC₅₀ = 862.60 nM). These promising results led us to test these six Ahp-cyclodepsipeptides against two other serine proteases, elastase and proteinase K. Elastase is produced in either the pancreas, for digestion of food, or by neutrophils for degradation of foreign proteins. Neutrophil elastase is a well-established drug target for the treatment of acute lung injury and acute respiratory distress syndrome[61]. Proteinase K is a fungal serine endopeptidase that is commonly used in molecular biology procedures[62]. This enzyme family plays important roles in the fungal infection of insects[63] and mammals[64]. While 2 was found to be a potent elastase inhibitor (IC₅₀ = 4.94 nM), 6 was discovered to be the most potent proteinase K inhibitor known to date (IC₅₀ = 5.42 nM). The isolated compounds were docked by induced-fit, inside the binding pocket of alpha-chymotrypsin (PDBID 4Q2K), and all were found to have a similar binding mode (Supplementary Methods and Supplementary Fig. 45) that was revealed by crystal structures of Ahp-cyclodepsipeptides in complex with serine proteases[65,66].

## Discussion

Here we describe the use of native metabolomics protein infusion MS to simultaneously detect protein-metabolite binding and annotate their molecular structures. This approach can be used for rapid screening of small molecule modulators for proteins of interest, directly from crude extracts. In our case study, we identified 30 chymotrypsin-binding natural products in a 10 min LC-MS run from a few µg of crude extract (not including downstream isolation and de novo structure elucidation). In comparison to flow-injection experiments (injection of crude extract in the native MS setup without

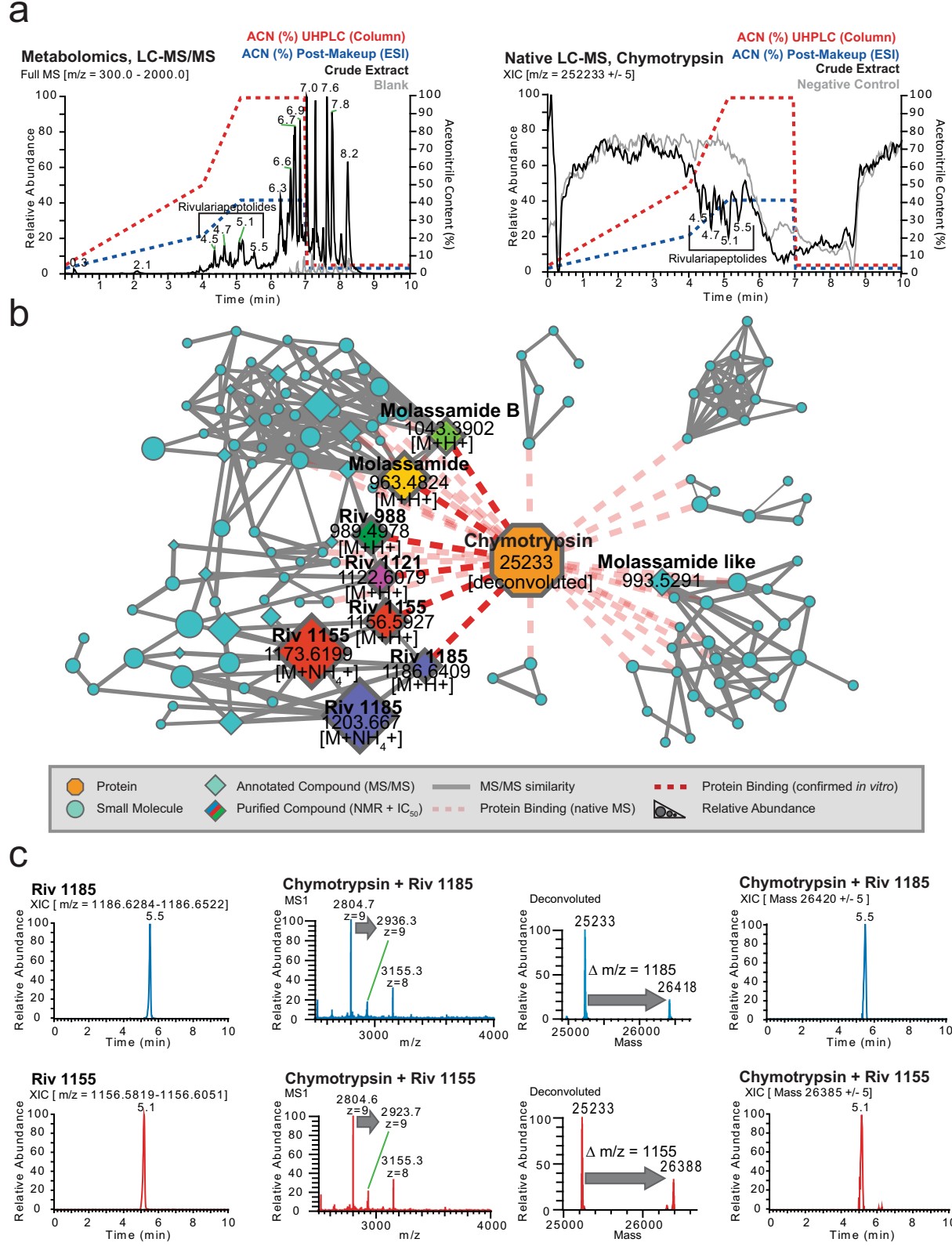

column), we observed a strong signal decrease, most likely through collective ion suppression effects (Supplementary Fig. 3d, left panel). When injecting preincubated chymotrypsin-crude extract mixture into our system (Supplementary Fig. 3d, right panel) we observed a more acceptable signal response and several protein-metabolite complexes, which we attributed to molassamide B, a molassamide derivative, and

rivulariapeptolide 1185. However, in comparison to native metabolomics, the number of putative binders observed was significantly lower, indicating that chromatographic separation is an important factor in the sensitivity of the approach. Most importantly, the chromatographic dimension is also essential for the unambiguous linking of binding information to MS/MS features which facilitates

**Fig. 2 | Native metabolomics analysis. a** Left panel: Total ion chromatogram (*m/z* 300–2000) of cyanobacterial crude extract and blank. The retention times (RT = 4.5–5.5 min) of major Ahp-cyclodepsipeptides are highlighted. Right panel: Deconvoluted extracted ion chromatogram (*m/z* 25,233 ± 5) of alpha-chymotrypsin screened against the cyanobacterial crude extract under native MS conditions and negative control. Acetonitrile (ACN) concentration on column and post-column (including make-up) are shown as dashed lines. The ACN concentrations are given at the pump for a given time, and a ~ 2 min delay time between pump and column has to be taken into account. **b** Correlation molecular network of deconvoluted chymotrypsin and putatively new small molecule inhibitors binders by native MS (A larger version of the network with detailed precursor mass labels of all nodes is available in the supporting information Supplementary Fig. 3). **c** Upper left panel: Extracted ion chromatogram (*m/z* 1,186.6284–1,186.6522) of putative chymotrypsin-binder rivulariapeptolide 1185 at RT = 5.5 min. Middle and upper right panel: Multiple charged and deconvoluted MS spectra and extracted ion chromatogram (*m/z* 26,420 ± 5) of a putative chymotrypsin-binder complex. The mass difference between the putative chymotrypsin-binder complex (*m/z* 26,420 ± 5) and apo-chymotrypsin (25,233 ± 5) suggests a molecular weight of 1187 ± 5 Da for the putative chymotrypsin-binder rivulariapeptolide 1185. Lower left panel: Extracted ion chromatogram (*m/z* 1,156.5819–1,156.6051) of putative chymotrypsin-binder rivulariapeptolide 1185 at RT = 5.1 min. Middle and lower right panel: Multiple charged and deconvoluted MS spectra Extracted ion chromatogram (*m/z* 26,385 ± 5) of a putative chymotrypsin-binder complex. Mass difference between the putative chymotrypsin-binder complex (*m/z* 26,385 ± 5) and apo-chymotrypsin (25,233 ± 5) suggests a molecular weight of 1152 ± 5 Da for the putative chymotrypsin-binder rivulariapeptolide 1155. (A larger version of mass spectra is shown in the supporting information Supplementary Fig. 5).

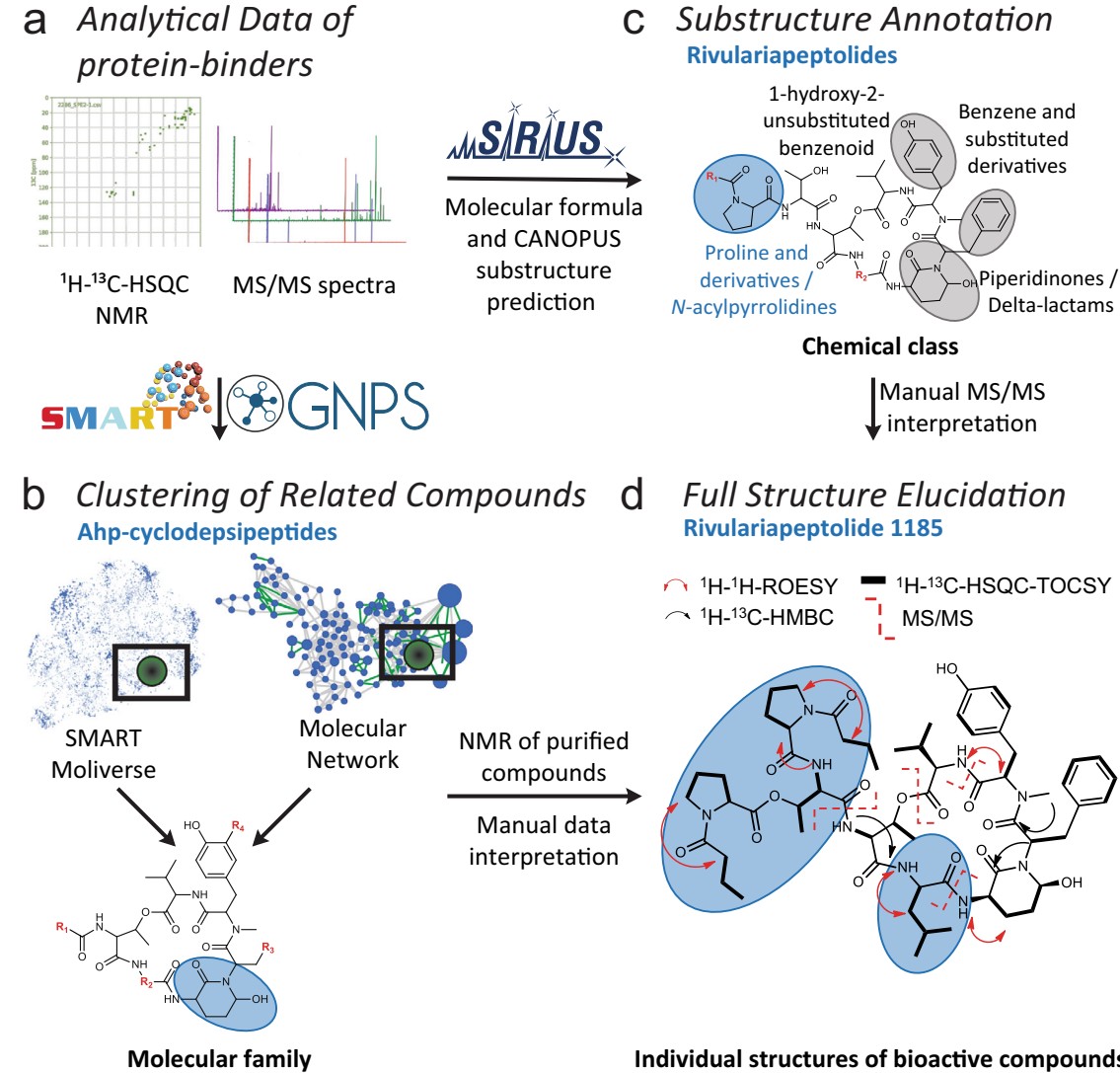

**Fig. 3 | Structure elucidation workflow based on NMR and MS/MS data.** a The workflow combined automated in-silico MS/MS and NMR annotation tools for fast compound class identification and dereplication of known natural products exemplified by the new natural product rivulariapeptolide 1185. **b** Molecular networking and SMART analysis suggested the presence of an Ahp-cyclodepsipeptide molecular family. **c** In-depth MS and MS/MS analysis of the new natural products with SIRIUS helped to establish the molecular formula and substructural information about the characteristic *N*-acylated proline residues. **d** Unambiguous structure elucidation by various 1D/2D NMR and MS/MS experiments led to the planar structure of rivulariapeptolide 1185. Selected 2D NMR-derived correlations and MS² fragmentations are depicted. The distinctive structural moieties are highlighted in blue bubbles and are described in the associated blue captions of panels **b**–**d**. Residues (R1–R4) highlighted in red represent unknown structural elements at the current analysis stage in panels **b**–**d**.

**Fig. 4 | Structure-activity relationships of rivulariapeptolides and molassamides. a** Structures of the isolated rivulariapeptolides 1185 (**1**), 1155 (**2**), 1121 (**3**), 988 (**4**); red = residues defined in panel **c**. **b** Structures of the isolated known molassamide (**5**), and the new molassamide B (**6**). **c** Potency of isolated compounds for selected serine proteases following 40 min pre-incubations. Data were presented as the mean ± SD, $n = 3$. Ba butyric acid, Pro proline. Source data are provided as a Source Data file.

dereplication and downstream structure elucidation. Online Post-column complex formation by addition of metal binding partner has been used, to study carbohydrates[67] and siderophores[68], respectively, which we here expand to protein targets as binding partners.

Initial tests with 12 proteins showed the accessibility of a wide range of protein targets by native metabolomics (Supplementary Fig. 46) as well as the need to optimize buffer concentrations and mass spectrometry settings. While this experiment highlights the accessibility of diverse proteins under native metabolomics conditions, we did not monitor ligand binding for these proteins. Since we did not assess binding properties with differentially affine ligands and proteins of different molecular weights, the suitability of our approach for studying protein-small molecule interactions of lower affinity and with larger proteins remains to be characterized. Still, we assume that native metabolomics is generalizable to other protein targets that are accessible via native ESI[69–71] provided that they are available in large enough quantities. The amounts of protein needed for native metabolomics high-throughput screening are a few mg for a 24 h screen of 96 samples with a 10 min LC-MS method, which is achievable for many commercially available proteins or those coming from in-house heterologous protein expression.

Besides the ad hoc experimental determination of binding information, we systematically organized this information in the public GNPS spectral library through spectrum tags. Hence, native metabolomics-derived properties are accessible in future experiments and can provide biological context to complex metabolomes.

We primarily used the method for an initial screening approach to assign binary binding information (binder/non-binder). In this study we did not determine absolute binding affinities by native metabolomics. Nevertheless, informed estimations can be made about relative binding affinities by fitting the intensity ratio of bound and unbound protein as a function of ligand concentration[21,72]. This method first assumes that no dissociation takes place during the transmission through the mass spectrometer, and second, that the observed gas-phase intensity ratio correlates with the solution ratio. To test the degree of in-source dissociation of protein-binder complexes using our native metabolomics approach, we varied the ESI parameters (in-source CID energy, HESI aux gas temperature, S-Lense RF Level) and observed the resulting change in protein-binder complex ratios compared to the apoprotein (Supplementary Fig. 47). We concluded that the protein-binder complex of interest does not undergo in-source dissociation over a wide range of parameter settings. Only very high CID levels (80 eV) or low S-Lense RF Levels (10%) can lead to the complete loss of signal for the protein-binder complex or low absolute intensity, respectively. These assumptions imply that ESI titration measurements can deliver a relative "snapshot" of the solution concentrations to reflect solution-phase binding affinities. Nevertheless, it is important to point out that the native metabolomics approach should be considered as a first hypothesis-generating tool, and that putative binders should be further validated with orthogonal methods to both determine absolute binding affinities, enzyme inhibition rates, and to out rule nonspecific binding effects.

From a natural product discovery perspective, it is very interesting that 30 putative bioactive molecules for a target protein were discovered from a single extract. This indicates that the chemical space for certain bioactive molecular families can often be underestimated

when compared to traditional bioactivity-guided approaches, as they are typically biased towards the most abundant or most active compounds. At least for Ahp-cyclodepsipeptides, recent biosynthetic studies suggest that the high structural diversity of these compounds is mainly driven by the hypervariability of amino acids in the positions proceeding and following the Ahp-unit (see Fig. 4a for the definitions of residues $R_3$ and $R_2$, respectively)[73]. The events impacting $R_2$ can be explained by high-frequency point mutations. This is sought to provide an evolutionary platform to iteratively test combinations while maintaining the central activity. However, the amino acid substitutions at $R_3$ most likely occur via recombination events, thereby allowing for evolutionary shortcuts[73]. These biosynthetic hypotheses are supported by the compounds isolated in this study and add to a better understanding of structure-activity relationships for Ahp-cyclodepsipeptides. Comparing rivulariapeptolide 1185 (**1**) to rivulariapeptolide 1121 (**3**), a leucine residue is swapped for an Abu unit at $R_2$, and phenylalanine is replaced by leucine at $R_3$. The most surprising structure-activity relationship gained from this study, however, was that a single substitution of bromine (molassamide B, **6**) for hydrogen (molassamide, **5**) led to a thirtyfive-, three-, and four-fold increase in potency towards chymotrypsin, elastase, and proteinase K, respectively.

The protease inhibition of the compounds discovered with the native metabolomics workflow was confirmed with an orthogonal fluorescence assay against three proteases. At nanomolar concentration, their $IC_{50}$ show high potency and exhibits distinct selectivity when compared to other Ahp-cyclodepsipeptides[36], Fig. 4 and Supplementary Table 5). For example, molassamide (**5**) is the second most potent inhibitor screened against proteinase K but is the least potent inhibitor for chymotrypsin and elastase. Rivulariapeptolide 1155 (**2**), on the other hand, is the most potent elastase inhibitor but shows much lower inhibition against both chymotrypsin and proteinase K.

Together, these findings highlight the utility of the native metabolomics approach presented herein. Beyond the discovery of protease inhibitors, we anticipate that native metabolomics may be applicable for the screening of a broad variety of interactions of biomolecules from complex mixtures at scale. We envision that native metabolomics might become a central tool for activity and functional-metabolomics workflows. This would not only benefit drug discovery and chemical ecology studies but could potentially also be leveraged for the generation of large-scale training data for machine learning approaches to predict protein-ligand interactions.

## Methods

### Cyanobacterial collection and taxonomy

Marine cyanobacteria biofilm samples were collected in an intertidal zone growing on rock/reef substrate near Las Palmas Beach, Manatí, Puerto Rico, U.S. (GPS coordinates: 18°28'32.0″N 66°30'00.5″W) on May 14th, 2019, and at 0.5–2.0 m of water at Carlos Rosario Beach in Culebra, Puerto Rico, U.S (GPS coordinates: 18°19'30.0″N 65°19'48.0″W) on April 6th, 2019. Biomass for both samples was hand collected (DRNA Permit O-VS-PVS15-SJ-01165-15102020). Microscopic examination indicated that this collection was morphologically consistent with the genus *Rivularia*. 16 S rDNA analysis confirmed the identity as *Rivularia* spp. PCC 7116. Voucher specimen available from E.C.D. as collection no. MAP14MAY19-1, and from W. H. G. as collection no. CUR6APR19-1.

### Micro-flow LC-MS/MS data acquisition

For micro-flow UHPLC-MS/MS analysis 2 μL were injected into vanquish UHPLC system coupled to a Q-Exactive (setup A) or a Q-Exactive HF (setup B) quadrupole orbitrap mass spectrometer (Thermo Fisher Scientific, Bremen, Germany) with an Agilent 1260 quaternary HPLC pump (Agilent, Santa Clara, USA) or in setup B a fully integrated

vanquish quaternary UHPLC pump (Thermo Fisher Scientific, Bremen, Germany) as make-up pumps. For reversed-phase chromatographic, a C18 core-shell micro-flow column (Kinetex C18, 150 × 1 mm, 1.8 um particle size, 100 A pore size, Phenomenex, Torrance, USA) was used. The mobile phase consisted of solvent A (H$_2$O + 0.1% formic acid (FA)) and solvent B (acetonitrile (ACN) + 0.1% FA). The flow rate was set to 150 μL/min (setup A) or 100 μL/min (setup B). In setup A, a linear gradient from 5–50% B between 0–4 min and 50–99% B between 4 and 5 min, followed by a 2 min washout phase at 99% B and a 3 min re-equilibration phase at 5% B. In setup B, a linear gradient from 5–50% B between 0-8 min and 50–99% B between 8 and 10 min, followed by a 3 min washout phase at 99% B and a 5 min re-equilibration phase at 5% B.

Metabolomics data-dependent acquisition (DDA) of MS/MS spectra was performed in positive mode. Electrospray ionization (ESI) parameters were set to 40 arbitrary units (arb. units) sheath gas flow, auxiliary gas flow was set to 10 arb. units and sweep gas flow was set to 0 AU. The auxiliary gas temperature was set to 400 °C. The spray voltage was set to 3.5 kV and the inlet capillary was heated to 320 °C. S-lens level was set to 70 V applied. MS scan range was set to 200–2000 $m/z$ with a resolution at $m/z$ 200 (R$_{m/z\ 200}$) of 70,000 with one micro-scan. The maximum ion injection time was set to 100 ms with automatic gain control (AGC) target of 5E5. Up to two MS/MS spectra per duty cycle were acquired at R$_{m/z\ 200}$ 17,000 with one micro-scan. The maximum ion injection time for MS/MS scans was set to 100 ms with an AGC target of 5.0E5 ions and a minimum of 5% AGC. The MS/MS precursor isolation window was set to $m/z$ 1. The normalized collision energy was stepped from 20 to 30 to 40% with z = 1 as the default charge state. MS/MS scans were triggered at the apex of chromatographic peaks within 2 to 15 s from their first occurrence. Dynamic precursor exclusion was set to 5 s. Ions with unassigned charge states were excluded from MS/MS acquisition, as well as isotope peaks.

For native metabolomics experiments, the same chromatographic parameters were used and in addition, 220 μL/min (setup A) or 150 μL/min (setup B) ammonium acetate buffer was infused post-column through a make-up pump and a PEEKT-splitter and enzyme solution was infused with 2 μL/min flow rate via the integrated syringe pump. ESI settings were set to 40 arb. units sheath gas flow, the auxiliary gas flow was set to 10 arb. units and sweep gas flow was set to 0 arb. units. The auxiliary gas temperature was set to 150 °C. The spray voltage was set to 3 kV and the inlet capillary was heated to 253 °C. S-lens level was set to 30 V applied. MS scan range was set to 2000-4000 $m/z$ (setup A) and 2500–4000 $m/z$ (setup B) with a resolution R$_{m/z\ 200}$ 140,000 (setup A) and 120,000 (setup B) with 2 microscans. MS acquisition was performed in all-ion fragmentation (AIF) mode with R$_{m/z\ 200}$ with 20% HCD collision energy and an isolation window of 2000 –4000 $m/z$ (setup A) or 2500–4000 $m/z$ (setup B).

### Native metabolomics data analysis

For native LC-MS data, multiple charged spectra were deconvoluted using the Xtract algorithm in Qualbrowser, part of the Xcalibur software (Thermo Scientific) and also using FLASHDeconv (version 2.0Beta) {Jeong, K. et al. FLASHDeconv: Ultrafast, High-Quality Feature Deconvolution for Top- Down Proteomics. Cell Syst 10, 213-218 e216 (2020).}. For Xtract algorithm, both deconvoluted native LC-MS and metabolomics LC-MS/MS.raw files were converted to the centroid.mzML file format using MSconvert of the proteowizard software package. For FLASHDeconv, the raw files were converted into centroid m/zML again using MSconvert software first. Then the converted mzML files were deconvolved to generate deconvolved.mzML format files. Feature finding of both file types was performed using a modified version of MZmine2.37 (corr.17.7). Feature tables from both intact protein mass and metabolomics data were matched by their retention time (RT) and an $m/z$ offset corresponding to the mass of chymotrypsin (25234 Da) with an RT tolerance of 0.2 min and a mass tolerance of 4 Da.

Feature tables (.csv), MS/MS spectra files (.mgf), and ion identity networking results (.csv) were exported and uploaded to the MassIVE repository. LC-MS/MS data were submitted to GNPS for feature-based molecular networking analysis. Downstream combined Molecular-Networks and chymotrypsin small molecule binding were visualized as networks in cytoscape (3.8.2).

## Optimization of ESI settings, buffer concentration, and pH dependency of native metabolomics

To optimize ESI setting as well as the influence of $NH_4Acetate$ concentration and pH, we used molassamide as a test compound. For optimization of ESI settings, we performed flow injections without a column and ramped ESI aux gas temperature, in-source CID energy, S-lense RF voltage. Optimization of pH values over the entire LC gradient and peak intensities were both assessed using three different make-up buffers. Make-up buffer 1 was water, make-up buffer 2 was 10 mM $NH_4Acetate$ buffer, and make-up buffer 3 was 10 mM $NH_4Acetate$ buffer + 0.2% ammonium hydroxide, v/v. Effects of $NH_4Acetate$ concentration was assessed with an online mixed make-up flow (using Experimental Setup B) with the low-pressure gradient pump. As solvent A we used LC-MS grade water and 100 mM $NH_4Acetate$ as solvent B. Molassamide was prepared as 100 μM solutions from a 10 mM stock solution in DMSO by preparing a 1:100 dilution into solvent mixture 1 (water + 10% acetonitrile + 0.1% formic acid). Samples were analyzed as described in micro-flow LC-MS/MS data acquisition (setup A); 2 μL of each solution were injected into the mass spectrometer, while chymotrypsin (Sigma), dissolved in water to a final concentration of 2 mg/mL, was injected through the syringe pump at a flow rate of 2 μL/min. pH values were assessed by disconnecting the flow to the source and collecting ~30 μL of solvent every minute, then testing this value on pH paper.

## Titration of ligands and concentration dependency

Molassamide was prepared as 100 μM solutions from a 10 mM stock solution in DMSO by preparing a 1:100 dilution into solvent mixture 1 (water + 10% acetonitrile + 0.1% formic acid). From this 100 μM solution, dilutions were prepared at 10, 1, and 0.1 μM into solvent mixture 1. 2 μL of each solution were injected into the mass spectrometer. Samples were analyzed as described in Micro-Flow LC-MS/MS data acquisition (setup A), while chymotrypsin (Sigma Aldrich) was dissolved in water to a final concentration of 2 mg/mL and injected through the syringe pump at a flow rate of 2 μL/min. The ratio of bound to unbound protein was plotted against the ligand concentration in a given HPLC peak window. Data points were fitted using the solver function.

## Determination of limit of detection, selectivity, and flow-injection experiments

Molassamide was dissolved in 50% MeOH and a serial dilution with a dilution factor of 2 and 10 were performed yielding final concentrations of 100, 50, 10, 1, and 0.1 μg/mL. Two microliters of each dilution were injected into the native metabolomics micro-flow LC-MS system (setup B) and peak areas from chymotrypsin-bound molassamide were extracted and plotted against their concentration. For testing the selectivity of the method, a series of standards (gallinamide A, FR900359, quinalizarin, genistein, phloroglucinol, cymodepsipeptide A, lingaoamide, molassamide B, rivulariapeptolide 1121, rivulariapeptolide 1185, and tutuilamide A) were dissolved to 100 μg/mL and pooled to final concentrations of 10 μg/mL in 50% MeOH. For flow-injection analysis, the 10 μg/mL molassamide standard was used and the UHPLC column was bypassed with a stainless steel union. Continuous injections during the micro-flow LC gradient were performed manually through the direct control function in the Xcalibur software (Thermo Scientific) with ~1 min spacing.

## Chymotrypsin activity assays in native mass spectrometry buffer

Cyanobacteria extract (1 mg/ml, methanol) was diluted in 10 mM ammonium acetate pH 4.5 to 30 ug/mL and then sequentially diluted 1.5-fold to 0.52 μg/mL in the same buffer. Bovine chymotrypsin (Sigma Aldrich) and Suc-Ala-Ala-Pro-Phe-AMC (Calbiochem, 230914) were diluted to 300 nM and 150 μM, respectively in 10 mM ammonium acetate pH 4.5. In a 384-well black microplate, 10 μL of enzyme, substrate, and cyanobacteria extract were combined (30 μL final volume) such that the concentrations in the reaction were 100 nM chymotrypsin, 50 μM of Suc-Ala-Ala-Pro-Phe-AMC and 10 μM to 0.17 nM of cyanobacteria extract. For solvent compatibility assays, chymotrypsin (100 nM) was assayed with 10 μg/mL and 1 μg/mL extract in 10 mM ammonium acetate, pH 4.5 containing 8.3 to 41.7% acetonitrile. A control assay lacked acetonitrile and cyanobacteria extract. All assays were performed in triplicate wells at 25 °C in a Synergy HTX Multi-Mode Microplate Reader (BioTek, Winooski, VT) with excitation and emission wavelengths of 360 and 460 nm, respectively. The initial reaction velocity in each well was recorded and the dose-response curve was generated using GraphPad Prism 9 software.

## Calculation of $IC_{50}$ values

Chymotrypsin (1 nM), proteinase K (10 nM), and elastase (20 nM) was preincubated with 0 to 3 μM of each compound for 40 min in Dulbecco's phosphate buffered saline, pH 7.4 containing 0.01% Tween-20. The reaction was initiated by the addition of 25 μM of Suc-Ala-Ala-Pro-Phe-AMC (Calbiochem, 230914) for proteinase K and chymotrypsin, 25 μM of MeOSuc-Ala-Ala-Pro-Val-AMC (Cayman, 14907) for elastase in a final volume of 30 μL, respectively. The release of the AMC fluorophore was recorded in a Synergy HTX multi-mode reader (BioTek Instruments, Winooski, VT) with excitation and emission wavelengths at 340 and 460 nm, respectively. The maximum velocity was calculated in RFU/sec over 10 sequential points on the linear part of the progress curve. The $IC_{50}$ values were determined by nonlinear regression in GraphPad Prism 9.

## NMR spectroscopy

Deuterated NMR solvents were purchased from Cambridge Isotope Laboratories. 1H NMR and 2D NMR spectra were collected on a Bruker Avance III DRX-600 NMR with a 1.7 mm dual-tune TCI cryoprobe (600 and 150 MHz for 1H and 13 C NMR, respectively) and a JEOL ECZ 500 NMR spectrometer equipped with a 3 mm inverse detection probe. NMR spectra were referenced to residual solvent DMSO signals ($\delta H$ 2.50 and $\delta C$ 39.5 as internal standards). The NMR spectra were processed using MestReNova (Mnova 12.0, Mestrelab Research) or TopSpin 3.0 (Bruker Biospin) software.

## Extraction and isolation

The preserved cyanobacterial biomass from collection no. CUR6APR19-1 was filtered through cheesecloth, and then (98.7 g dry wt) was extracted repeatedly by soaking in 500 mL of 2:1 $CH_2Cl_2$/MeOH with warming (<30 °C) for 30 min to afford 1.44 g of dried extract. A portion of the extract was fractionated by reversed-phase solid phase extraction ($C_{18}$-SPE) using a stepwise gradient solvent system of decreasing polarity (Fr. 1–1 35% ACN/$H_2O$, 124.4 mg; Fr. 1–2 70% ACN/$H_2O$, 76.1 mg; Fr. 1-3 100% ACN, 77 mg; Fr. 1–4 100% MeOH, 254.9 mg. Fr. 1–2 was dissolved in 70% ACN/H2O and purified by preparative HPLC using a Kinetex 5 μm RP 100 Å column (21.00 × 150 mm) and isocratic elution using 50% ACN/H2O for 8 min, then ramping up to 100% in 14 min at the flow rate of 20 mL/min, yielding 56 subfractions. Rivulariapeptolide 1185 (compound **1**) and 1155 (compound **2**) were isolated from subfractions 1-2-20 to 1-2-23 that were combined (4.5 mg) and further purified by semi-preparative HPLC using a Synergi 4 μm Hydro-RP 80 Å column (10.00 × 250 mm) and isocratic elution gradient

elution using 35% ACN/65% $H_2O$ isocratic at the flow rate of 3.5 mL/min for 3 min the ramping up to 55% ACN in 22 min, then ramping up to 100% ACN in 1 min and holding the gradient at 100% ACN for another 5 min yielding **1** (1.6 mg, RT = 23.3 min) and **2** (1.3 mg, RT = 21.8 min) as a colorless, amorphous solid. The same HPLC conditions were used to isolate compounds **3** (rivulariapeptolide 1121, from subfraction 1-2-10 and 1-2-11, 1.1 mg, RT = 18.5 min) and **4** (rivulariapeptolide 988, from subfraction 1-2-7, 1.3 mg, RT = 12.6 min). Fr. 1-1 was dissolved in 30% ACN/H2O and purified by preparative HPLC using a Kinetex 5 µm RP 100 Å column (21.00 × 150 mm) and isocratic elution using 30% ACN/H2O for 10 min, then ramping up to 50% in 10 min and then to 95% in 2 min at the flow rate of 20 mL/min, yielding 29 subfractions. Molassamide (compound **5**) was isolated from subfractions 1-1-6 and 2-1-10 were combined (3.3 mg) and further purified by semi-preparative HPLC using a Synergi 4 µm Hydro-RP 80 Å column (10.00 × 250 mm) and isocratic elution gradient elution using 35% ACN/65% H2O isocratic at the flow rate of 3.5 mL/min for 3 min the ramping up to 55% ACN in 22 min, then ramping up to 100% ACN in 1 min and holding the gradient at 100% ACN for another 5 min yielding **5** as a colorless, amorphous solid (1.8 mg) at RT = 10.8 min. The same HPLC conditions were used to isolate compound **6** (Molassamide B, from subfraction 1-1-4 and 1-1-5, 1.8 mg, RT = 13.7 min).

### Reporting summary

Further information on research design is available in the Nature Research Reporting Summary linked to this article.

### Data availability

All raw (.raw), deconvoluted (xtract.raw), and centroided (.mzXML or.mzML) mass spectrometry data as well as processed data feature table (.csv) and MS/MS spectra (.mgf) are available through the MassIVE repository (massive.ucsd.edu) with the identifier MSV000087964 [https://doi.org/10.25345/C5G25F], MSV000088586 [https://doi.org/10.25345/C5N585], and MSV000088578 [https://doi.org/10.25345/C5NW1R]. The MS/MS spectra of the newly described derivatives, including tags as protease inhibitors, have been added to the GNPS library (gnps.ucsd.edu) with the following IDs: rivulariapeptolide 1185 (**1**): CCMSLIB00005723387; rivulariapeptolide 1155 (**2**): CCMSLIB00005723986, CCMSLIB00005720236; rivulariapeptolide 1121 (**3**): CCMSLIB00005723398; rivulariapeptolide 988 (**4**): CCMSLIB00005723393; molassamide (**5**): CCMSLIB00005723404; molassamide B (**6**): CCMSLIB00006710020. Raw NMR data and structures for compounds **1** -**6** have been deposited to Zenodo at https://doi.org/10.5281/zenodo.6956292. Structural analysis of rivulariapeptolides peptides bound to α-chymotrypsin was based on the previously published Protein Data Bank entries 4GVU and 4Q2K. Source data are provided with this paper.

### Code availability

The modified version of MZmine2.37 (corr.17.7) used in this study is available at https://github.com/robinschmid/mzmine2/releases. The code for the mass-offset matching for native metabolomics data analysis is available under https://github.com/Functional-Metabolomics-Lab/Native-Metabolomics and https://doi.org/10.5281/zenodo.6786930. FLASHDeconv is a part of OpenMS and is available as platform independent open-source software under a BSD three-clause license at https://OpenMS.org/FLASHDeconv.

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

## Acknowledgements

We thank the Deutsche Forschungsgemeinschaft for the support of D.P. through a postdoctoral research fellowship (PE 2600/1-1) and of D.P., P.S.1, B.W., K.S., P.S.2, H.B.-O., and C.C.H. through the CMFI Cluster of Excellence (EXC 2124, project-IDs 390838134, and 1-06.010_0) and the Collaborative Research Center CellMap (TRR 261, project-ID 398967434). R.R., P.C.D., and W.H.G. were supported by the Gordon and Betty Moore Foundation (GBMF7622) and by the US National Institutes of Health (R01 GM107550, P41 GM103484, and R03 CA211211). W.B. was supported in part by the Research Foundation—Flanders (12W0418N). We thank the Spanish Ministry of Education and Science for their support of A.I.P.-L. through the program FPU (FPU19/00289), of C.M.-S. through the program Juan de la Cierva-Incorporación (IJC2018-036923-I) and D.R. (PID2019-107724GB-I00). D.R. was supported by the European Research Council with a Starting Grant (BacBio 637971) and the Proyectos I + D + I Program PAIDI 2020 of Junta de Andalucía (P20_00479).

## Author contributions

R.R., and D.P. conceived the study. W.H.G. supervised the fieldwork. R.R., K.L.A., C.B.N., E.J.C., and W.H.G. collected and extracted environmental samples. A.T.A. and D.P. developed the native metabolomics approach. A.I.P.-L., J.H.W.S., P.M.M.H., W.E.D., C.M.-S., D.R., K.S., P.S., H.B.-O., and D.P. provided proteins. W.B. and K.J. wrote the software. M.W. aided in integration with GNPS tags. R.R., A.T.A., P.S., B.W., and D.P. performed MS experiments. R.R. and M.L.M-H. performed compound isolation. R.R. carried out NMR experiments. R.R., P.S., C.C.H., and D.P. performed total hydrolysis and derivatization experiments. P.F., C.L., and A.J.O. performed activity assays. I.Y.B.S. performed docking studies. P.C.D., W.H.G., and D.P. acquired funding and provided infrastructure. R.R. and D.P. wrote the manuscript. All authors edited and approved the final manuscript.

## Funding

## Competing interests

P.C.D. and W.H.G. are scientific advisors of Sirenas. P.C.D. is a scientific advisor of Galileo, Cybele, and scientific advisor and co-founder of Ometa Labs LLC and Enveda with approval by the UC San Diego. M.W. is the founder of Ometa Labs LLC. The remaining authors declare no competing interests.
