## [Peer Review File · Nature Communications]

REVIEWER COMMENTS

Reviewer #3 (Remarks to the Author):

Overall, the authors have nicely addressed my concerns and those of the two other reviewers. In particular, the additional discussion added to the manuscript makes a much better case for the novelty of the approach by placing it in perspective concerning other techniques. Compared to some of those other methods, the strengths of the method are then well-illustrated now by some of the additional experimental detail. My minor concern regarding the configurational assignment of the new compounds has been addressed. I would recommend publishing with very minor corrections.

In the intervening time since the original submission, the authors have published another paper in Nature Chemistry (doi.org/10.1038/s41557-021-00803-1) titled Native mass spectrometry-based metabolomics identifies metal-binding compounds. For several reasons, I cannot access the full article while writing this review. Still, it appears the experimental set-up is essentially the same as is the general goal of identifying binders from complex mixtures. I did not find a reference to this published paper in the revised draft. While some mention of that paper should be included, I do not feel that this prior publication negatively impacts the claims of novelty in the revised manuscript. Demonstrating the utility of the method for screening as outlined in the revised manuscript is distinct enough, in my opinion, in case the other reviewers raise this concern.

Two other very minor corrections I would suggest are:

1) Several discussions of the general SAR trends of Ahp-containing compounds have been published. Including a discussion in this paper with reference would be helpful for the reader to place these changes in context.

1) Figure 3 caption: "The Unique structural moieties" – suggest replacing "unique" with "distinct" for accuracy.

Reviewer #4 (Remarks to the Author):

Petras and coworkers present a new native metabolomics approach to enable bioactivity-focused compound identification from complex metabolite mixtures. Proposed approach involves non-targeted LC-MS/MS and simultaneous detection of protein binding via native LC-MS (native metabolomics LC-MS setup). Expected and new interactions were detected (not quantified) and some of them were confirmed by another (biochemical) assay. While efficient screening assays for target-based natural product drug discovery are needed, they need be shown to be reliable and robust. To ensure assay reliability, typically a wide range of model protein-ligand interactions are used. However, in this paper validation of the method shown only for interactions of a single protein with very strong binders. Also parts of the manuscript related to native LC-MS raise concerns. These are described below.

1) The absence of discussion (and references) of relevant literature pertaining to the use of native MS to screen natural libraries suggests that the authors not adequately familiar with the field. There are multiple published MS-based methods in JACS, 2021, 143, 21379-21387; Anal Chem. 2020,92:14012-14020; Proc. Natl. Acad. Sci. U.S.A. 2000, 97, 12008; J Biol Chem. 2005,280,32200; Analyst, 2018, 143, 536 for natural library screening. On-line post-column complex formation by addition binding partner was also shown previously in Anal. Chem. 1995, 67, 3501.

2) The authors should clearly say that the presented method does not allow K_d determination. The text from line 352 is misleading because it gives the impression that the proposed method can determine K_d . No titration plots are shown in the paper and according to described data, initial concentrations of protein target and ligands are not known.

Lines 352-355: "While we primarily used the method for an initial screening approach and assigned binary binding information (binder/non-binder), titration experiments using native mass spectrometry can be used to determine relative dissociation constants (K_d) by fitting the intensity ratio of bound and unbound protein as a function of the added ligand.

3) Lines 355-360: "This method first assumes that no dissociation takes place during the transmission through the mass spectrometer, and second, that the observed gas phase intensity ratio correlates

with the solution ratio. These assumptions imply that ESI titration measurements can deliver a relative “snapshot” of the solution concentrations to reflect solution-phase binding affinities.”

The authors provide no evidence to support their claim that in-source dissociation does not occur. High flow rates and harsh source conditions needed for desolvation in native LC-MS may lead to full or partial protein-ligand complex dissociation, especially for gas-phase labile complexes (not be confused with weak solution interactions).

4) Lines 360-362: “Nevertheless, as the experimental environment is inherently different (gas phase vs. solution) absolute binding affinities might differ from solution based orthogonal assays.” This statement should be removed because “experimental environment” in the proposed method for binding events is solution, not gas-phase. However, solution conditions for binding are not characterized properly (no description of time-dependent protein and ligand concentrations, solution temperature, time-dependent solvent composition).

5) The occurrence of nonspecific binding of small molecules to the protein and protein-ligand complex (or complexes) is not discussed at all. In fact, there is no single mass spectrum of detected protein and protein-ligand complex. Such spectra have to be shown for each time window corresponding to individual metabolite components. Were ions of the protein target bound to more than one ligand molecule detected? If yes, it will indicate nonspecific binding of the ligands to the target protein.

6) Lines 87-89: “An important inherent limitation in the use of ligand pools is that multiple ligands compete for binding of the target at the same site, and therefore compounds with highest affinity or concentration are more easily discovered.” I don’t agree with this statement. Library screening by native MS is typically done with excess of binding sites (on target protein) and therefore all ligands have access to free binding sites.

7) Authors need to demonstrate the use of the method using different (larger and heterogeneous) protein targets and lower affinity interactions.

8) Another feature of the method that can be prohibitive for many interactions is that protein-ligand interactions have to occur in solution with high organic solvent concentration and at acidic pH. Authors need to comment on what metabolites interactions with protein targets can tolerate these conditions.

9) 10mM AmAc is quite low salt concentration. Native MS binding assay is typically performed at ~200mM AmAc. What is largest salt concentration that can be used (considering possible salt precipitation when mixed with ACN and likely ESI instability at high salt content)?

10) In figure caption for Figure 4 "(b) Potency..." should be "(c) Potency..."

Point-to-point reply

REVIEWER COMMENTS

Reviewer #3 (Remarks to the Author):

Overall, the authors have nicely addressed my concerns and those of the two other reviewers. In particular, the additional discussion added to the manuscript makes a much better case for the novelty of the approach by placing it in perspective concerning other techniques. Compared to some of those other methods, the strengths of the method are then well-illustrated now by some of the additional experimental detail. My minor concern regarding the configurational assignment of the new compounds has been addressed. I would recommend publishing with very minor corrections.

In the intervening time since the original submission, the authors have published another paper in Nature Chemistry (doi.org/10.1038/s41557-021-00803-1) titled Native mass spectrometry-based metabolomics identifies metal-binding compounds. For several reasons, I cannot access the full article while writing this review. Still, it appears the experimental set-up is essentially the same as is the general goal of identifying binders from complex mixtures. I did not find a reference to this published paper in the revised draft. While some mention of that paper should be included, I do not feel that this prior publication negatively impacts the claims of novelty in the revised manuscript. Demonstrating the utility of the method for screening as outlined in the revised manuscript is distinct enough, in my opinion, in case the other reviewers raise this concern.

Thank you very much for your time reviewing our manuscript and for the positive feedback and for your suggestions. We agree, this is of course an important paper to cite. When we originally submitted the manuscript, the paper was still under review and we mistakenly did not include it during the first revision stage. We now reference this paper (Nature Chemistry, doi.org/10.1038/s41557-021-00803-1, Ref⁶⁷) and other post-column strategies (Leary et al. <https://doi.org/10.1021/ac00115a019>, Ref⁶⁶) in the discussion.

Quote text (lines 377-382):

“Online Post-column complex formation by addition of metal binding partner has been used, to study carbohydrates⁶⁶ and siderophores⁶⁷, respectively, which we here expand to protein targets as binding partners. Initial tests with 12 proteins showed the accessibility of a wide range of protein targets by native metabolomics (Figure S46) as well as the need to optimize buffer concentrations and mass spectrometry settings.”

Two other very minor corrections I would suggest are:

1) Several discussions of the general SAR trends of Ahp-containing compounds have been published. Including a discussion in this paper with reference would be helpful for the reader to place these changes in context.

Thank you for the suggestion. We added the following sentence to the discussion section:

Quote text

l. 435-436:

“At nanomolar concentration their IC50 show high potency and exhibit distinct selectivity when compared to other Ahp-cyclodepsipeptides³⁶, Figure 4, Table S4).”

1) Figure 3 caption: “The Unique structural moieties” – suggest replacing “unique” with “distinct” for accuracy.

Thank you, we implemented the change as suggested.

Reviewer #4 (Remarks to the Author):

Petras and coworkers present a new native metabolomics approach to enable bioactivity-focused compound identification from complex metabolite mixtures. Proposed approach involves non-targeted LC-MS/MS and simultaneous detection of protein binding via native LC-MS (native metabolomics LC-MS setup). Expected and new interactions were detected (not quantified) and some of them were confirmed by another (biochemical) assay. While efficient screening assays for target-based natural product drug discovery are needed, they need be shown to be reliable and robust. To ensure assay reliability, typically a wide range of model protein-ligand interactions are used. However, in this paper validation of the method shown only for interactions of a single protein with very strong binders. Also parts of the manuscript related to native LC-MS raise concerns. These are described below.

1) The absence of discussion (and references) of relevant literature pertaining to the use of native MS to screen natural libraries suggests that the authors not adequately familiar with the field. There are multiple published MS-based methods in JACS, 2021, 143, 21379-21387; Anal Chem. 2020,92:14012-14020; Proc. Natl. Acad. Sci. U.S.A. 2000, 97, 12008; J Biol Chem. 2005,280,32200; Analyst, 2018, 143, 536 for natural library screening. On-line post-column complex formation by addition binding partner was also shown previously in Anal. Chem. 1995, 67, 3501.

Thank you very much for your time reviewing our manuscript and we appreciate very much your suggestions to improve the paper. The references that were recommended have now been added to our discussion section. At the time of the original submission some of these papers had not yet been published and we were unaware of the others. We agree with the reviewer that they are relevant to our work and thus included them following paragraphs:

Quote text:

I. 86-89:

“Native electrospray ionization (ESI) such as immobilized enzyme MS (IEMS)¹⁵ and affinity mass spectrometry (MS) such as pulsed ultrafiltration (UF) MS, size exclusion (SEC) or affinity bead-based pull-down assays are increasingly being used to analyze non-covalent binding of biomolecules¹⁶⁻²¹.”

I. 94-96

“Both native and affinity MS approaches have been applied with single compounds as well as substrate pools, which allows for the simultaneous screening of complex natural compound libraries²⁶⁻²⁸.”

I. 377-382:

“Online Post-column complex formation by addition of metal binding partner has been used, to study carbohydrates⁶⁶ and siderophores⁶⁷, respectively, which we here expand to protein targets as binding partners. Initial tests with 12 proteins showed the accessibility of a wide range of protein targets by native metabolomics (Figure S46) as well as the need to optimize buffer concentrations and mass spectrometry settings.”

2) The authors should clearly say that the presented method does not allow K_d determination. The text from line 352 is misleading because it gives the impression that the proposed method can determine K_d. No titration plots are shown in the paper and according to described data, initial concentrations of protein target and ligands are not known.

Lines 352-355: “While we primarily used the method for an initial screening approach and assigned binary binding information (binder/non-binder), titration experiments using native mass spectrometry can be used to determine relative dissociation constants (K_d) by fitting the intensity ratio of bound and unbound protein as a function of the added ligand.

We agree with the reviewer that the presented method currently does not allow for accurate and absolute K_d determination, and thus rephrased our statement accordingly.

At the same time, we would like to emphasize that the presented native metabolomics approach is developed for scalable prioritization and annotation of new protein binders, and the determination of binding affinities was not in the focus of our work. Nevertheless, native mass spectrometry has been used to determine binding affinities and we show a concentration dependent titration plot for the protein/binder pair chymotrypsin/molassamide (Figure S1c), which we hope is better reflected now in our modified statement.

Quote text:

I.392-396:

“We primarily used the method for an initial screening approach to assign binary binding information (binder/non-binder). In this study we did not determine absolute binding affinities by native metabolomics. Nevertheless, informed estimations can be made about relative binding affinities by fitting the intensity ratio of bound and unbound protein as a function of ligand concentration^{21,72}.”

3) Lines 355-360: “This method first assumes that no dissociation takes place during the transmission through the mass spectrometer, and second, that the observed gas phase intensity ratio correlates with the solution ratio. These assumptions imply that ESI titration measurements can deliver a relative “snapshot” of the solution concentrations to reflect solution-phase binding affinities.”

The authors provide no evidence to support their claim that in-source dissociation does not occur. High flow rates and harsh source conditions needed for desolvation in native LC-MS may lead to full or partial protein-ligand complex dissociation, especially for gas-phase labile complexes (not be confused with weak solution interactions).

This is an excellent point, thank you very much. While there are several studies demonstrating that binding affinities derived from native ESI-MS data correlate well to results of complementary assays (e.g. ITC, Su et al. <https://doi.org/10.1038/s41401-020-0483-6>) we are not aware of studies or approaches to experimentally investigate absolute in-source dissociation.

To test the degree of relative in-source dissociation of protein binder complexes using our native metabolomics approach, we performed a set of experiments that varied in our opinion critical ESI parameters such as in-source CID energy, HESI aux gas temperature, S-Lense RF level. Our data indicated that, the relative abundance of protein-ligand complex remains stable over a broad range for HESI Aux gas temperature (heated Nitrogen) and S-Lense RF level (Entry ion transfer optics in the Q Exactive) and only observed significant signal decline at low S-Lense RF Levels and high in-source CID energies (Figure S47). From these results, we hypothesize that the protein-ligand complex does not undergo significant in-source dissociation over a wide range of parameter settings. We added the new data to the supplementary information and added the following statement to the main text.

Quote text:

I. 398-412:

“To test the degree of in-source dissociation of protein binder complexes using our native metabolomics approach, we varied the ESI parameters (in-source CID energy, HESI aux gas temperature, S-Lense RF Level) and observed the resulting change in protein-binder complex ratios compared to the apoprotein (Figure S47). We concluded that the protein-binder complex of interest does not undergo in-source dissociation over a wide range of parameter settings. Only extremely high CID levels (80 eV) or low S-Lense RF Levels (10%) can lead to the complete loss of signal for the protein-binder complex or low absolute intensity, respectively. These assumptions imply that ESI titration measurements can deliver a relative “snapshot” of the solution concentrations to reflect solution-phase binding affinities. Nevertheless, it is important to point out that the native metabolomics approach should be considered as a first hypothesis generating tool, and that putative binders should be further validated with orthogonal methods to both determine absolute binding affinities, enzyme inhibition rates, and to outrule non-specific binding effects.”

Figure S47: Assessment of native electrospray and in-source dissociation conditions for chymotrypsin and molassamide. Flowinjections of molassamide were performed in triplicates for each condition. a) shows the XICs of the chymotrypsin x molassamide complex (top left) and apo chymotrypsin (bottom left) over stepwise ramped in source CID energy (1-80 eV). The scatter-plot on the right shows the relative peak height (apo vs. holo) over the different CID energy. b) displays the XICs of the chymotrypsin x molassamide complex (top left) and apo chymotrypsin (bottom left) over stepwise increased HESI AUX temperature (40-

300 °C). The scatter-plot on the right shows the relative peak height (apo vs. holo) over the different temperature steps. c) displays the XICs of the chymotrypsin x molassamide complex (top left) and apo chymotrypsin (bottom left) over stepwise increased relative S-lense radiofrequency (RF) level (10-90%). The scatter-plot on the right shows the relative peak height (apo vs. holo) over the different RF levels.

4) Lines 360-362: “Nevertheless, as the experimental environment is inherently different (gas phase vs. solution) absolute binding affinities might differ from solution based orthogonal assays.” This statement should be removed because “experimental environment” in the proposed method for binding events is solution, not gas-phase. However, solution conditions for binding are not characterized properly (no description of time-dependent protein and ligand concentrations, solution temperature, time-dependent solvent composition).

Thanks. We agree with the reviewer and removed this statement.

5) The occurrence of nonspecific binding of small molecules to the protein and protein-ligand complex (or complexes) is not discussed at all. In fact, there is no single mass spectrum of detected protein and protein-ligand complex. Such spectra have to be shown for each time window corresponding to individual metabolite components. Were ions of the protein target bound to more than one ligand molecule detected? If yes, it will indicate nonspecific binding of the ligands to the target protein.

Thanks a lot for the comment. We agree with the reviewer, non-specific binding can be a limitation in native MS. We are aware of this, and similar as in other studies, we used a baseline collision energy to mitigate eventual non-specific binding. To point this out more clearly and state that native metabolomics should be considered as a hypothesis generating tool and that ultimately orthogonal determination of absolute binding affinities and inhibition rates are needed, we added new sentences in the results and the discussion section.

Quote Text:

I. 159-161:

“To reduce non-specific binding, native metabolomics experiments were performed in all-ion fragmentation (AIF) mode, using a threshold collision energy of 20 % in the HCD cell.”

I. 408-412:

“Nevertheless, it is important to point out that the native metabolomics approach should be considered as a first hypothesis generating tool, and that putative binders should be further validated with orthogonal methods to both determine absolute binding affinities, enzyme inhibition rates, and to outrule non-specific binding effects.”

With regards to the binding of multiple ligands, this is a great suggestion, although it does not exclude that a protein might have multiple (allosteric) binding sides. Interestingly, we observed low abundant double bond species for molassamide, a known nanomolar chymotrypsin binder. We added a short statement on this as well.

I. 177-181:

“We also observed a minor peak with mass shift that corresponds to two molasamide molecules in the multiple charged spectrum (Figure S1d) which can be an indication of non-specific binding⁵¹. However, as molassamide is a known nanomolar inhibitor, we can not rule out whether the second peak could be due to non-specific binding effects or if it represents a secondary weaker interaction.”

We added now the full native MS spectra for all new inhibitors discovered and discussed in the text in Figure 2, Figure S1d, and Figure S5.

to be taken into account. (b) Correlation molecular network of deconvoluted chymotrypsin and putatively new small molecule inhibitors binders by native MS (A larger version of the network with detailed precursor mass labels of all nodes is available in the supporting information Figure S3). (c) Upper left panel: Extracted ion chromatogram (m/z 1,186.6284 - 1,186.6522) of putative chymotrypsin-binder rivulariapeptolide 1185 at RT = 5.5 min. Middle and upper right panel: Multiple charged and deconvoluted MS spectra and extracted ion chromatogram (m/z 26,420 \pm 5) of a putative chymotrypsin-binder complex. Mass difference between the putative chymotrypsin-binder complex (m/z 26,420 \pm 5) and apo-chymotrypsin (25,233 \pm 5) suggests a molecular weight of 1,187 \pm 5 Da for the putative chymotrypsin binder rivulariapeptolide 1155. Lower left panel: Extracted ion chromatogram (m/z 1,156.5819 - 1,156.6051) of putative chymotrypsin-binder rivulariapeptolide 1185 at RT = 5.1 min. Middle and lower right panel: Multiple charged and deconvoluted MS spectra and extracted ion chromatogram (m/z 26,385 \pm 5) of a putative chymotrypsin-binder complex. Mass difference between the putative chymotrypsin-binder complex (m/z 26,385 \pm 5) and apo-chymotrypsin (25,233 \pm 5) suggests a molecular weight of 1,152 \pm 5 Da for the putative chymotrypsin binder rivulariapeptolide 1155. (A larger version of mass spectra is shown in the supporting information Figure S5).

Figure S1: Optimization of native MS conditions a) Integrated peak area of the protein-ligand complex increases in a pH-dependent manner with increasing pH. (b) Molassamide was screened against chymotrypsin as a proof-of-concept experiment. A $\Delta m/z$ of 962.5 Da, the difference between unbound chymotrypsin and the chymotrypsin-molassamide complex, corresponds to the m/z of molassamide. (c)

Concentration-dependent increase in chymotrypsin-molassamide complex, measured by native metabolomics. Concentration_{Peak} (μM) refers to the concentration of the ligand. (d) Mass spectrum from chymotrypsin-molassamide complex from native metabolomics run. (e) The limit of detection (LOD) for the molassamide-chymotrypsin interaction was determined by generating a calibration curve for molassamide and injecting molassamide stock solutions of 0.1 – 100 $\mu\text{g}/\text{mL}$ and chymotrypsin. The extracted ion chromatograms (XICs) for the chymotrypsin-molassamide complex are shown. (f) Concentration-dependent inhibition of chymotrypsin activity by 10 $\mu\text{g}/\text{mL}$ of cyanobacterial crude extract in 10 mM ammonium acetate pH 4.5. (g) Relative chymotrypsin activity of cyanobacterial crude extract (1 $\mu\text{g}/\text{mL}$ and 10 $\mu\text{g}/\text{mL}$) in native MS conditions (10 mM ammonium acetate pH 4.5 plus increasing concentrations of acetonitrile) compared with no addition of crude extract.

Chymotrypsin x Rivulariapeptolide 988

Chymotrypsin x Rivulariapeptolide 1121

Chymotrypsin x Rivulariapeptolide 1155

Chymotrypsin x Rivulariapeptolide 1185

Figure S5: Mass spectra from chymotrypsin-rivulariapeptolide complexes from native metabolomics runs.

6) Lines 87-89: “An important inherent limitation in the use of ligand pools is that multiple ligands compete for binding of the target at the same site, and therefore compounds with highest affinity or concentration are more easily discovered.” I don’t agree with this statement. Library screening by native MS is typically done with excess of binding sites (on target protein) and therefore all ligands have access to free binding sites.

Thank you for your comment. And we agree with the reviewer, some studies indeed paid particular attention to perform the incubation with molar excess of the protein, while other studies used moderate to extreme molar excess of ligands (see table below).

To differentiate better between this, we modified our statement as follows:

Quote text:

I. 96-101:

“Depending on the experimental setup, an important limitation in the use of ligand pools is that multiple ligands compete for binding of the target at the same site, and therefore compounds with highest affinity or concentration are more easily discovered. However, this limitation can be overcome by using a molar excess of protein compared with the total molar concentration of the ligand library¹⁵.”

Review Table 1. Molar ratios of target protein and ligand pools

Paper	Concentration Protein	Concentration Ligand Pool (average MW ~ 500 Da)	Protein / Ligand Ratio	Reference
Nguyen et al. JACS 2021	20 μ M	5 mg/mL (~ 10000 μ M)	0.002	https://pubs.acs.org/doi/10.1021/jacs.1c10408
Van Breemen et al. JNP 2022	50 pM	0.1 μ M (MW CBDA = 358.478 g/mol)	0.0005	https://pubs.acs.org/doi/10.1021/acs.jnatprod.1c00946
Cancilla et al. PNAS 2000	150 nM	38 nM	3.95	https://www.pnas.org/doi/10.1073/pnas.220403997
Yu et al. JBC 2005	40 μ M	200 μ M	0.2	https://www.jbc.org/article/S0021-9258(20)79193-3/fulltext
El-Hawiet et al. Analyst 2018	5 μ M BabA	0.05 mg/mL fractions ((50 μ M) for average MW = 1000 g/mol)	0.1	https://pubs.rsc.org/en/content/articlelanding/2018/an/c7an01397c
	15 μ M hGal-3C	0.05 mg/mL fractions ((50 μ M) for average MW = 1000 g/mol)	0.3	
Park et al. Anal. Chem. 2020	5 μ M SNA	10 nM N-Glycan library	500	https://pubs.acs.org/doi/10.1021/acs.analchem.0c02931
	8 μ M MAA	10 nM N-Glycan library	800	
	6 μ M hCD22-Fc	100 nM N-Glycan library	60	
Bui et al. Anal. Chem. 2022	10 μ M CD22	0.5 mg/mL ((200 μ M) for average MW = 2500 g/mol)	0.05	https://pubs.acs.org/doi/abs/10.1021/acs.analchem.1c04779
	8 μ M MAA	0.08 mg/mL ((32 μ M) for average MW = 2500 g/mol)	0.25	

7) Authors need to demonstrate the use of the method using different (larger and heterogeneous) protein targets and lower affinity interactions.

Thanks a lot for the suggestion. We agree with the reviewer that showing the accessibility of different protein targets will strengthen our work and support our claim of generalizability. While we do have only limited accessibility of different protein-ligand pairs, and / or those are currently subject of ongoing studies, we now included results from 12 additional proteins that showcase stable ionizability over the LC gradients which we deem most critical for native metabolomics.

From initially 14 tested proteins, 12 gave a stable signal over at least the majority of the LC gradient. The proteins included monomeric and multimeric proteins of diverse classes and molecular weights (14kDa – 80kDa, Figure S46). While most proteins tolerate acetonitrile concentrations over the full gradient (1%-41.7% ACN) others were more susceptible to higher contents of organic solvents as expected.

The two proteins that did not work in our approach were highly glycosylated and are generally challenging in native MS measurements (e.g. IgG).

We included the new data as a figure in the supplemental information and discussed the results in the main text.

Quote text:

I. 380-384:

“Initial tests with 12 proteins showed the accessibility of a wide range of protein targets by native metabolomics (Figure S46) as well as the need to optimize buffer concentrations and mass spectrometry settings. We thus assume that native metabolomics is generalizable to other protein targets that are accessible via native ESI⁶⁸⁻⁷⁰ and which are available in large enough quantities.”

Figure S46: Assessment of accessibility of different protein targets for native metabolomics. Extracted ion chromatograms and Full mass spectrum of target proteins under native MS conditions. Acetonitrile (ACN) concentration on column and post-column (including make-up) are shown as dashed lines. The ACN concentrations are given at pump for a given time and a ~ 2 min delay time between pump and column has to be taken into account. Mobile phase NH₄Ac concentrations are indicated in the headings for each

protein. (a) BSA (bovine serum albumin, *Bos taurus*, pdb id 4F5S). (b) Carbonic anhydrase (*Bos taurus*, pdb ID 1V9E). (c) Elastase (*Sus domesticus*, pdb id 1INC). (d) Lysozyme (*Gallus domesticus*, pdb id 5YIN). (e) Proteinase K (*Tritirachium album*, pdb id 2PRK). (f) Transferrin *Homo sapiens* pdb id 6CTC). (g) Chitosanase (*Bacillus subtilis*, pdb id 7C6C). (h) CutA (Divalent-cation tolerance protein CutA. *Escherichia coli*) pdb id 3X3U, (i) NS2B-NS3 protease (Dengue NS2B-NS3 protease, *Dengue virus*, pdb id 4M9I). (j) FtsZ (Cell division protein FtsZ, *Staphylococcus aureus*, pdb id).(k) Pim1 (Serine/threonine-protein kinase pim-1, *Homo sapiens*, pdb id 1YXT). (l) TasA (TasA protein, *Bacillus subtilis*, pdb id 5OF1).

8) Another feature of the method that can be prohibitive for many interactions is that protein-ligand interactions have to occur in solution with high organic solvent concentration and at acidic pH. Authors need to comment on what metabolites interactions with protein targets can tolerate these conditions.

Thanks a lot, and we agree that the organic solvent content can be a limitation for some protein-ligand interactions as ESI ionization efficiency can significantly vary or some proteins might denature at high ACN contents. However, in our setup the maximum ACN concentration is 41.7 %, because of our make-up flow dilution post-column which can also be further lowered by increasing the make-up flow rate.

To avoid misunderstandings, we plotted the ACN concentration (on column and post-make-up) in the XIC in Figure 2a and Figure S2a. In Figure S46 we could show that all 12 proteins that were tested tolerated ACN contents of at least 30% ACN in the post-makeup flow, corresponding to 15% ACN on the UHPLC gradient; many proteins were even stable over the whole gradient.

Volatile ammonium acetate buffer is the most commonly used non-denaturing solvent in native MS measurements of proteins. Even though having initially neutral pH ammonium acetate has no buffering capacity at pH 7 and in positive ion mode there are several factors that tend to acidify the analyte solution. In the presence of ammonium acetate, the pH may drop to values as low as 4.75 ± 1 in the final ESI droplets, reflecting the pKa of acetate buffer (Konermann et al. 10.1007/s13361-017-1739-3).

To make this clearer in the manuscript we included the latter paragraph in the main text.

Quote text:

I. 167-172:

“Volatile ammonium acetate buffer is the most commonly used non-denaturing solvent in native MS measurements of proteins. Even though having initially neutral pH ammonium acetate has no buffering capacity at pH 7 and in positive ion mode there are several factors that tend to acidify the analyte solution. In the presence of ammonium acetate, the pH may drop to values as low as 4.75 ± 1 in the final ESI droplets, reflecting the pKa of acetate buffer⁵⁰.”

9) 10mM AmAc is quite low salt concentration. Native MS binding assay is typically performed at ~200mM AmAc. What is largest salt concentration that can be used (considering possible salt precipitation when mixed with ACN and likely ESI instability at high salt content)?

Thanks a lot and we agree with the reviewer, the NH₄Acetate concentration can have a large influence on the ionization and eventual dissociation constants of noncovalent protein-binder complexes). In our case we initially tested a series of NH₄Acetate concentrations (10mM - 100mM) and found 10 mM to be the most favorable to study chymotrypsin-ligand complexes as we observed highest relative abundance, highest chymotrypsin-ligand to chymotrypsin ratio and the lowest abundance of interfering NH₄Acetate clusters. However, as the reviewer will see, in the new data from the other tested proteins (Figure S46) the concentrations we used ranged from 10-100 mM. To point this out in the manuscript, we added a short paragraph as follows and added the figure comparing different buffer concentrations in the SI (figure S2).

Quote text:

I. 201-208:

“The NH₄Acetate concentration can have a large influence on the ionization and dissociation constants of noncovalent protein-binder complexes. Interestingly, higher ammonium acetate concentration can lead to both higher or lower binding affinities depending on the protein-ligand pair investigated⁵². We tested a series of NH₄Acetate concentrations (10mM - 100mM, Figure S2) and found 10 mM to be the most favorable to study chymotrypsin-molassamide complexes (highest relative abundance, highest chymotrypsin-molassamide to chymotrypsin ratio, lowest abundance of interfering NH₄Acetate clusters).”

Figure S2: Optimization of NH₄Acetate buffer concentration. Native LC-MS analysis with molassamide and chymotrypsin were performed under different make-up buffer concentrations (10-100 mM final) to determine optimal binding/ MS sensitivity. a) shows the extracted ion chromatograms of the chymotrypsin x molassamide complex. b) shows the average MS spectrum (8.4-8.6 min) of the chymotrypsin x molassamide complex. c) displays a reference MS spectrum of chymotrypsin at the max. ACN (40%) concentration at the end of the gradient.

10) In figure caption for Figure 4 “(b) Potency...” should be “(c) Potency...”

Thanks a lot. Fixed as suggested.